# A Fault-Detection System Approach for the Optimization of Warship Equipment Replacement Parts Based on Operation Parameters

**DOI:** 10.3390/s23073389

**Published:** 2023-03-23

**Authors:** Álvaro Michelena, Víctor López, Francisco Lamas López, Elena Arce, José Mendoza García, Andrés Suárez-García, Guillermo García Espinosa, José-Luis Calvo-Rolle, Héctor Quintián

**Affiliations:** 1Department of Industrial Engineering, University of A Coruña (UDC), CTC, CITIC, Rúa Mendizábal, s/n, 15403 Ferrol, Spain; elena.arce@udc.es (E.A.); hector.quintian@udc.es (H.Q.); 2Centro de Supervisión y Análisis de Datos de la Armada (CESADAR), Arsenal de Cartagena, Armada Calle Real s/n, 30290 Cartagena, Spain; flamlop@mde.es (F.L.L.); jmendoza@isdefe.es (J.M.G.); ggespinosa@isdefe.es (G.G.E.); 3Computing and Artificial Intelligence Laboratory (CAILab), Facultad de Ciencia y Tecnología, Universidad Camilo José Cela, Calle Castillo de Alarcón 49, 28692 Madrid, Spain; 4Área de Sostenimiento y Gestión Logística, ISDEFE, Calle Beatriz de Bobadilla, 3., 28040 Madrid, Spain; 5Spanish Naval School, University Defense Center, 36920 Marín, Spain; andres.suarez@cud.uvigo.es

**Keywords:** fault detection, one-class, warship, machine learning

## Abstract

Systems engineering plays a key role in the naval sector, focusing on how to design, integrate, and manage complex systems throughout their life cycle; it is therefore difficult to conceive functional warships without it. To this end, specialized information systems for logistical support and the sustainability of material solutions are essential to ensure proper provisioning and to know the operational status of the frigate. However, based on an architecture composed of a set of logistics applications, this information system may require highly qualified operators with a deep knowledge of the behavior of onboard systems to manage it properly. In this regard, failure detection systems have been postulated as one of the main cutting-edge methods to address the challenge, employing intelligent techniques for observing anomalies in the normal behavior of systems without the need for expert knowledge. In this paper, the study is concerned to the scope of the Spanish navy, where a complex information system structure is responsible for ensuring the correct maintenance and provisioning of the vessels. In such context, we hereby suggest a comparison between different one-class techniques, such as statistical models, geometric boundaries, or dimensional reduction to face anomaly detection in specific subsystems of a warship, with the prospect of applying it to the whole ship.

## 1. Introduction

Among many benefits, there are two significant cross-cutting advantages in ship automation: the crew diminution and the hazards and risks reduction of the people in general and the ship itself [1]. The first implies a significant reduction in ship operating costs directly [2]. Indirectly, it would entail aspects related to improving the conditions of workers in general terms (training, social security, the possibility of oversizing the workforce, …) [3]. On the second way, remark that in general terms, automation confers advantages with reducing risks for workers since it is the automatic machine that performs the work and the worker normally reduces his work to supervision. In addition, tedious and repetitive tasks, with the problems that these entails, are carried out by automata [3].

From a technical point of view, automation implies optimization if the accomplishing is correct [4]. In this sense, if the ship processes are well automated, then as result, advantages such as emissions reduction, consumption reduction, increased reliability, and availability could be achieved. Consequently, aspects as important as sustainability and reduction of environmental impact would be favored [5]. These aspects must continue to be pursued and improved as technology advances, and as there is a margin within which to do so.

The current convulsed geopolitical situation invites us to pay special attention to the military sector.Given the evolution of today’s societies, losing lives in armed conflicts is tremendously unpopular from a country’s political point of view [6]. Because of all this, it is a priority right now to put the minimum number of human lives at risk. It is especially important to try to move towards solutions similar to those of unmanned aerial vehicles (UAVs) or functional robots. The government’s goal is to achieve zero human deaths in the face of any military intervention [6].

In the specific case of warships, all the ship aspects mentioned before are applicable. However, it is necessary to consider and add the characteristics and conditions of military ships. In addition to multiple singularities, this kind of ship usually has more confined and optimized spaces [7]. Due to these characteristics, it would be desirable for this type of ship to transport only the essentials, and of course to guarantee satisfactory operation.

In general terms, the ships have a certain useful life. The same thing happens with warships, and it is even more restrictive. The fact that the useful life is shorter will depend, among others, on the economy of the country, of popularity from a social point of view, and on obsolescence from a military point of view [8]. Operationally, ships should ideally be at the forefront of technology.

Like in other sectors, the design of the warships obviously depends on the current technology and its future trend. Nowadays, as in the industry, there is a very strong trend in the naval sector towards digitization [9]. From an operational point of view, warships tend to install this type of technology because of the advantages they bring [10]. Thanks to this kind of relatively incipient technology, the optimization possibilities on ships are huge compared to the current state. It is necessary to emphasize that due to issues such as climate change or global warming, it is also mandatory that everything is carried out from a point of view of sustainability, energy efficiency, and low emissions.

Anomaly or failure detection is one of the objectives that must be achieved in any sector [11]. Digitization allows and helps carrying out this kind of technique, even giving new advantages from a reliability and accuracy point of view [12]. If this task is carried out satisfactorily, new advantages can be achieved in operation. Some of them are, for example, the increase in the efficiency of maintenance tasks, shipment of the optimal number of spare parts, or the optimization of the material purchase process.

There are many anomaly detection methods [13,14]. Consequently, there are several possible classifications that can be made of the types of existing techniques [13]. In some cases, an exhaustive knowledge of the system on which failure detection is intended is necessary [15]. This fact can have certain advantages during the process, such as better results, accuracy, and/or performance [12]. However, in cases where operating personnel change frequently, it can be very important that exhaustive process knowledge is not necessary [12]. Thus, they could perform anomaly detection autonomously or with non-expert operators.

Anomaly detection in several ships’ systems is essential in order to know the operational status of the frigate. Therefore, analyzing and detecting anomalies or malfunctions is a previous and necessary step to predict the possible spare parts to be shipped depending on the ship’s condition.

This research paper deals with a novel fault detection system approach applied to certain onboard subsystems, aiming to expand to the entire ship. In this manner, our proposal tests a set of different intelligent techniques following a one-class conceptualization, where only information from the system’s normal operation is consumed to model its performance during the training stage.

The present research work is organized as follows. After the current introduction, the related works are described. After this section, the case of the study is explained, detailing the equipment and the information system structure. Then, the materials and the methods taken into account for the present research are detailed. The work is continued with a description of the experiments and their results. Finally, conclusions and future works are presented.

## 2. Related Work

Throughout this section, we briefly mention recent work that may be related to the present study, either according to the techniques employed or the application to warship equipment.

Many reviews are oriented toward maintenance practices and strategies. In this sense, Kimera et al. [16] provide an overview of the maintenance parameters and practices that are critical for marine mechanical systems classified as plant, machinery, and equipment (PME). Previously, Cullum et al. [17] focused on the risk-based maintenance (RBM) scheduling framework as applied to warships and naval vessels, and provided a critical analysis of the risk assessment and maintenance scheduling techniques in use. More recently, Zhang et al. [18] reviewed the methods, strategies, and application of marine systems and equipment (MSAE) in prognostics and health management (PHM) as an essential means to optimize resource allocation and improve the intelligent operation and maintenance (O&M) efficiency of MSAE.

Along these lines, in relation to the PHM of naval mechanical systems, there are other works from the perspective of technical processes. For example, online condition monitoring and self-repair techniques for in-service marine diesel engines [19], different types of blade failures and current blade failure-detection methods [20], and mainly data-driven models and the problem of condition-based maintenance of marine propulsion systems are reviewed [21]. In the field of autonomous shipping, Karatug et al. [22] propose an evaluation of PHM systems and reliability centered maintenance strategies, as one method that can be implemented to cover the three major elements of maintenance management systems: risk assessment, maintenance strategy selection, and maintenance task interval, according to Emovon et al. [23] for autonomous marine vessels.

For its part, considering the current status and future trends of maintenance strategies applied in particular to corroded marine steel structures, Abbas et al. [24] analyzed deterministic and probabilistic models for predicting corrosion rates.

Other works propose their own strategies, suggesting a predictive maintenance solution based on a computational artificial intelligence model using real-time monitoring data, as stated by Jimenez et al. [25]. They also expose a ship-level method for repair decisions based on the preventive maintenance concept, relying on an improved failure mode and effects analysis (FMEA) method along with a Weibull distribution model, where the parameters are intended to be calculated using the maximum likelihood estimation (MLE) to predict the characteristic life of the equipment, and then determine the actions to be taken regarding logical decision principles and rule-based reasoning (RBR) in agreement with Song et al. [26].

A short time ago, in search of tackling decision making problems, Emovon et al. [27] presented a comparison of hybrid multi-criteria decision making (MCDM) methods for the selection of appropriate maintenance strategies for ship machinery systems and other related ship systems. Similarly, employing dynamic condition monitoring and historic data to present decision support information onboard, Michala et al. [28] proposed a novel decision support system (DSS) beforehand. Lately, making use of subjective opinions in DSS, Maurice et al. [29] attempted a technique for order preference by similarity to an ideal situation for ranking the maintenance strategies. Prior studies have proven that a decision support mechanism needs multiple criteria decision on ship equipment maintenance strategy selection [30,31,32].

With the specific aim of minimizing maintenance costs and maximizing ship reliability simultaneously, Zhao et al. [33] tested whether a bi-objective model under a condition-based maintenance strategy was applicable to the point of being able to provide support for ship operators.

In the arena of machine learning (ML) and especially fault detection systems, quite a few works can be found in the literature. Studies alike employ several prediction techniques, including regularized linear regression methods such as L1 (lasso) and L2 (ridge), or long short-term memory (LSTM)-based networks [34]. In this context, previous research has used artificial neuronal networks (ANN) [35] and decision trees [36], as well as support vector machines (SVM) [37,38] or the k-nearest neighbor technique (k-NN) as one of the most common for fault classification [39]. Moreover, in this scope, various deep learning (DL) techniques have also been used, as in the case of autoencoder neural networks [40] or recurrent neural networks (RNN) [41] based on long short-term memory network (LSTM) cells according to Yang et al. [42].

Some of the latest research achievements on the DL-based state of health prognostic methods address issues with limited labeled samples without assuming that the training and test datasets come from identical machines operating under similar conditions. In this context, Zhu et al. [43] developed Bayesian semi-supervised transfer learning with an active querying-based intelligent fault prognostic framework for remaining useful life (RUL) prediction across completely different machines under limited data.

As a compilation of the state of the art, recent reviews presented developments within fault detection and diagnosis (FDD) methods [44], introduced a survey of trends and techniques of fault detection in networked dynamical systems (NDS) [45], and gave an overview the theory and strategies of transient fault detection [46].

Other review papers lead to a specific field of application, as may be the optimal performance of photovoltaic (PV) systems [47,48], uninterrupted and trouble-free operation of induction motors (IMs) [49] or transient stability in a transmission network in agreement with Mishra et al. [50].

Working on novel approaches, some studies are available within the industrial processes area, proposing a fault detection system based on intelligent techniques [51,52]. In addition, certain previous works particularly focused on alternatives to hardware-only and software-only systems, as mentioned by Reis et al. [53] who identifed hybrid hardware/software fault-detection mechanisms.

To conclude, several references based on the techniques that stand the anomaly detection approach in the current study are present in the bibliography and discussed in more detail across Section 4.

## 3. Case of Study

This section introduces the technical–administrative processes, as far as the study is concerned, through which logistical support and sustainability considerations of material solutions are integrated into the navy as we know it today, specifically for the Spanish. The context in which the project takes place and where all the data monitoring and management is performed by the Data Supervision and Analysis Center of the Spanish Navy (CESADAR).

### 3.1. Equipment

Generally speaking, warships are very expensive systems, extraordinarily complex, with a lengthy procurement acquisition process and long operational life. During their life cycle, their configuration changes as a result of repairs and modifications. Therefore, the objective of integrated logistics support (ILS) [54] is to maximize system availability by optimizing the life cycle cost and ensuring mission compliance.

Another essential process is logistics configuration (LC), also referred to as the configuration, which encompasses the logistics processes required for thedevelopment of the ILS during the life cycle, defining the systems, subsystems, equipment, and components installed, with a description of their functional, physical and technical characteristics, quantity, type, and number of modifications made to each of them, as well as all available supporting documentation, recorded in documents and databases.

Following a functional criterion, the configuration is represented by a AEC (configured elements tree) that has a hierarchical structure. In the AEC, each element is assigned a HSC (hierarchical structure code) that works as a functional marking, up to 12 digits, which makes it possible to identify and distinguish that element univocally from others. Thereby, this HSC code is used as a reference in efforts such as the maintenance processes, spare parts management, etc.

By way of example, Figure 1 depicts how the AEC is developed on the basis of the operational requirements, identifying the functional nodes in accordance with the navy’s material guidelines, which defines the levels of hierarchical decomposition for assigning the functional mark of an element in the AEC.

Thereby, in order to control the evolution of onboard equipment, systems engineering has as its most effective tool the configuration management (CM) system, which provides an overview of the system development. At the same time, it is also an indispensable aspect for achieving interoperability, mitigating risks, and making effective use of NATO (The North Atlantic Treaty Organization) capabilities [56] in joint operations.

For the achievement of CM, the Spanish navy, through CESADAR, is supported by information and communication technologies (ICT) using a range of logistics applications which are itemized in the following subsection.

### 3.2. Warship Information System Structure

With participation in a multitude of missions since their commissioning in 2002, mainly under the NATO umbrella, the Spanish Álvaro de Bazán class frigates (F-100) were appointed to the investigation. Playing the role of the main source of information, CESADAR is the centralized system in charge of receiving, checking, and analyzing the data recorded in the vessels, and which in turn is stored and structured within its data lake.

Up until now, CESADAR has as one of its main applications the automation of surveillance and analysis tasks logistics application (ATAVIA), which provide a system for detecting anomalies in the operation of the monitored equipment based on expert rules. Indeed, all sensorized onboard systems and equipment are analyzed.

In addition to these, several separate applications developed in different programming languages and databases have been employed so far for logistics management, mainly from maintenance and procurement, posing a considerably complex architecture [55].

These logistics applications are difficult systems that require in-depth knowledge and expert analysis. Likewise, some authors have reviewed these tools proposing an alternative semi-unsupervised predictive maintenance system [34]. From our side, we focus on a novel fault detection system approach based on operational data provided by the ATAVIA application, as discussed below.

## 4. Materials and Methods

This section describes the techniques considered to achieve anomaly detection. The datasets used are also presented.

### 4.1. Employed Methods

The proposal follows a one-class approach, in which only information from the normal operation of the system is used to model its performance. Then, the detection of anomalous situations must be addressed without prior knowledge of their nature.

According to the literature [57], the implementation of one-class techniques can be based on three different principles: dimensional reduction techniques, statistical models, and the determination of geometric boundaries. To assess the performance of these three different approaches, four different techniques are proposed to model the normal behavior of the ship to detect anomalies.

#### 4.1.1. Statistical Models

An interesting approach to face anomaly detection using one-class techniques is based on the idea of using density functions. One commonly used statistical model is tested in this work.

##### Gaussian Model

One of the most direct ways to achieve a one-class classifier consists of applying a Gaussian distribution function over the training set, also known as a target set [57]. The covariance matrix and the mean vector are calculated and then a new test value is labeled depending on the score achieved at the Gaussian function.

Let us suppose a test instance 
$x\in R^{n}$
, whose distribution probability function is described in Equation (Equation 1). The function would produce lower values if *x* is not part of the target class. Therefore, selecting an appropriate threshold value would enable correct classification.

(1)
$$p\left(x,\mu ,\Sigma \right)=\frac{1}{{\left(2\pi \right)}^{n/2}{\left|\Sigma \right|}^{1/2}}e^{-\frac{1}{2}{\left(x-\mu \right)}^{T}\Sigma^{-1}\left(x-\mu \right)}$$

where:
μ
 is the training set mean value;
Σ
 is the training set covariance matrix.

This simple idea is characterized by a low computational cost, with the calculation of the covariance matrix being the most challenging step. Incorporating a regularization parameter 
$R_{p}$
 can be a valuable tool, particularly in cases where the inverse of the covariance cannot be computed.

Although this method demonstrates excellent performance, particularly when the target set is normally distributed [58], it does share a limitation with other density estimation techniques: the requirement for a sufficiently large training dataset [59].

#### 4.1.2. Geometric Boundaries

The calculation of the geometric boundaries of the target set can represent a good approximation to determine the difference between expected and unexpected events.

##### K-Means

The k-means unsupervised algorithm represents an intuitive way to achieve a one-class classifier, depending on the geometric distribution of the data [60].

After selecting the desired number of groups, the k-means algorithm partitions a given dataset into k clusters by minimizing the objective function expressed in Equation (Equation 2).

(2)
$$e=\sum_{k=1}^{C}\sum_{x\epsilon \varrho_{k}}^{}{∥x-c_{k}∥}^{2}$$

where:*x* represents a new input vector;
$c_{k}$
 denotes the centroid of the *k* cluster.

The centroids for each group are calculated using the training set to use k-means as a one-class technique. The distance of a given test data point to its closest centroid is then compared to the distances between each cluster data point and its respective centroid. The test data point is classified as anomalous if the distance exceeds all such distances.

For example, Figure 2 illustrates a case where the training set is divided into two clusters. In this instance, the black dot, representing a test point, is classified as a target because its distance to the nearest centroid (the orange star of Cluster 2) is lower than the distances of most training samples.

#### 4.1.3. Dimensional Reduction

The last one-class approach consists of the application of dimensional reduction techniques to learn patterns that only are presented in the target set.

##### Autoencoder

Utilizing artificial neural networks (ANN) configured with an autoencoder design in dimensional reduction techniques has produced substantial favorable outcomes [61].

To this end, the multilayer perceptron (MLP), which is one of the most common ANN architectures, is implemented. The MLP typically comprises an input layer, an output layer, and one or more hidden layers. The neurons in the contiguous layers are connected by weighted links that are automatically adjusted to minimize the discrepancy between the produced output and the desired output. Each layer also possesses an activation function applied to all the neurons within that layer, such as the linear, step, tan-sigmoid, or log-sigmoid function. The hidden layer output is computed using Equation (Equation 3),

(3)
$$h_{i}=f_{1}\left(W_{1}x_{i}+b_{1}\right)$$

where:
$h_{i}$
 defines the output of the hidden layer;
$f_{1}$
 is the hidden layer activation function;
$W_{1}$
 corresponds to the weight matrix between input and hidden layer;
$x_{i}$
 is the input vector;
$b_{1}$
 denotes the bias vector.

Subsequently, the output of the ANN is obtained using Equation (Equation 4).

(4)
$$\hat{x}=f_{o}\left(W_{o}h_{i}+p_{o}\right)$$

where:
$\hat{x}$
 is the ANN output.
$f_{o}$
 denotes the activation function of the output layer.
$W_{o}$
 define the weight matrix between hidden and output layers.
$p_{o}$
 is the bias vector.

The fundamental principle of the autoencoder involves training an ANN so that the output 
$\hat{x}$
 is equal to the input *x* while also executing a nonlinear reduction within the hidden layer through the activation function. This means that the number of neurons in the hidden layer must be lower than that of the inputs, which results in decompression being performed at the output [61]. The autoencoder method operates on the premise that anomalous points are significantly different from standard points in the hidden layer subspace, and the decompression process leads to high reconstruction errors.

##### Principal Component Analysis

Principal component analysis (PCA) is a commonly used technique for data dimensionality reduction tasks. This technique aims to identify the directions in the data with the highest variability and use them to establish new variables [60,62]. These directions are referred to as components and are calculated using the eigenvectors obtained from the eigenvalues of the covariance matrix.

PCA projects the original data points into the eigenvectors subspace, resulting in linear transformations. This technique is particularly effective in cases where the subspace is clearly linear.

In addition to its performance in dimensional reduction tasks, PCA can also be applied to one-class problems using reconstruction error criteria. For instance, suppose we have a training set *X*

$\in R^{2}$
 and use one of the two principal components to linearly transform it into 
$\hat{X}$

$\in R^{1}$
. A test data point 
$x_{t}$
 is labeled based on a reconstruction error criteria (Equation (Equation 5)), calculated as the difference between the initial point 
$x_{t}$
 and its projection 
$\hat{x_{t}}$
.

Anomalous points are likely to have higher reconstruction errors, and hence when the reconstruction error of a test data point is above a specific threshold, the anomaly is detected.

(5)
$$e\left(x\right)=\left|\right|x_{t}-{\hat{x}}_{t}\left|\right|$$


An instance where the distance from a test point to its projection exceeds all distances from the training points to their respective projections is illustrated in Figure 3. In this scenario, only the first component is utilized. The threshold distinguishing normal from anomalous behavior is frequently associated with the percentile distance of the training set.

### 4.2. Dataset

The information available through the abovementioned application, ATAVIA, has been used to obtain the working datasets. This application includes sensorization, warnings, and alarms of malfunctioning systems and equipment. For the present research, different subsystems of the vessels were selected as the object of analysis, selecting diesel power generation equipment. Specifically, three different datasets corresponding to the operation of a different components of the diesel generator have been used. No information about the study component and dataset features is provided for confidentiality reasons. In addition, generic names will be used to refer to them.

The working datasets contain the information captured by the sensors associated with each component within a 1 min period (sample rate of 0.016667 Hz), as well as the target variable, which indicates whether the sample corresponds to the normal or anomalous system performance. The operating data are available from May 2011 to May 2022.

It is important to emphasize that in this research only data recorded with the diesel generator operating in a stationary regime have been used, since the fault detection system from which the data are obtained generates a large number of false alarms in the system start-up and shutdown processes, minimizing the dataset quality.

Finally, the three working datasets are:Dataset 1 contains two variables and a total of 902,796 samples, of which 219 correspond to anomalous data.Dataset 2 contains two variables and 897,191 samples, of which 101 correspond to anomalous data.Dataset 3 contains twenty-five variables and 887,294 samples, of which 233 correspond to anomalous data.

## 5. Experiments Description (Setup) and Results

The research presented in this paper discusses the performance of three alternative one-class approaches. For this purpose, four different techniques are evaluated to model the normal performance of selected components of the power-generation system of the F-100 frigates. To obtain the best performance of each technique, several experiments have been carried out.

### 5.1. Experimental Setup and Assessment

To validate and compare the proposed techniques, different experiments have been carried out for each of the three datasets mentioned in Section 4.2.   To achieve the best classifier, the performance of each technique has been evaluated for different values of its hyperparameters. The evaluated configurations are listed below.
Gaussian Model−Data normalization−Data regularization−Outlier factorK-means−Data normalization−Number of clusters−Outlier factorAutoencoder−Data normalization−Neurons in the hidden layer−Outlier factorPrincipal Component Analysis−Data normalization−Number of components considered−Outlier factor

Each technique has been evaluated for three different data conditions. First, the data were used without any normalization process, *NoNorm*, then the data were normalized using a 0 to 1 criterion, *Norm*, and finally, the z-score method was applied (with a mean of 0 and a standard deviation of 1), *Zscore* [63]. The use of these normalization techniques, *Zscore*, in combination with one-class classification methods, has shown satisfactory results [64].

Moreover, each technique has been tested with different percentages of outliers in the training dataset. Values of 0, 5, 10, and 15% were used. The different configurations that have been tested for each technique are summarized in Table 1. All the one-class techniques have been implemented in *Matlab R2022b* using different packages and toolboxes.

Finally, it is important to highlight that k-fold cross-validation with a k = 3 was used for model validation and training. A schematic of the experimental setup is shown in Figure 4.

### 5.2. Results

The classifier performance evaluation has been measured using the area under the receiver operating curve (AUC) metric. This metric is well known in classifier performance analysis, and its result, measured as a percentage, relates false positive and true positive rates. In addition, the AUC is commonly used in cases where the dataset is unbalanced, as it is insensitive to class imbalance. Since the k-fold cross-validation method is used, the AUC results correspond to the mean performance of each configuration. In this way, each classifier’s training time has been considered to evaluate the computational cost. As a matter of fact, the results obtained for the different configurations of each technique are shown in Table 2, Table 3, Table 4, Table 5, Table 6 and Table 7.

Table 2 collects the results for the different configurations evaluated with the Gaussian model. As shown, in the three datasets the best AUC scores are registered with the data without normalization, with a regularization value of 0.009 and an outlier factor of 5% (in dataset 1 and 2) and 15% (in dataset 3). With this particular combination, with the first dataset, an AUC of 95.797% was obtained, 97.592% with the second data set, and 86.222% with the high-dimensional dataset.

In general terms, it can be observed that the Gaussian model is a technique with a low computational cost. In fact, none of the tested configurations exceeded a training time of 0.01 s for both datasets with two features, and for the third dataset, most settings had training times of less than 1 s.

Similarly, the resolution for the different sets of parameters evaluated with the k-means technique are summarized in Table 3. In this case, other combinations provide better performance for each dataset. As depicted, with dataset 1, the most favorable outcomes are obtained with a total of 2 clusters, an outlier factor of 10%, and normalizing the data with the z-score method; achieving with this specific configuration a 95.075% AUC. On the other hand, in the second experiment, with dataset 2, a percentage of 97.544% AUC was scored with the 6-cluster classifier implemented with an outlier factor of 5%, and the data normalized with the 0/1 criterion. Finally, the best classifier for the third dataset is obtained with 4 clusters, an outlier factor of 10%, and a Zscore normalization. With this configuration, an AUC value of 91.539% is achieved.

Generally speaking, with the k-means technique, good efficiency is appreciated with a fairly low computational cost. Still, analyzing Table 3, it is noticeable how the classifiers with an outlier factor of 0% do not achieve beneficial values. This fact can be produced as a consequence of possible anomalies labeled as normal data.

As before, Table 4 and Table 5 introduce the findings validated with PCA. Table 4 shows the results for the first two datasets of 2 variables, while Table 5 shows the performance of PCA with the dataset of 25 variables. It is important to note that Table 5 only offers the best configurations obtained by selecting each of the different numbers of the components since if all the tested configurations were shown, the table would contain a large amount of information that may compromise its comprehensibility.

In this case, analyzing Table 4, the most effective design is similar for both datasets since the best behavior is displayed with one component and the data normalized with the 0/1 criterion. The difference in the configurations corresponds to the percentage of the outlier factor. Firstly, through simulation with dataset 1, the highest AUC value, 88%, is collected with an outlier factor of 15%, while in the second case, the best classifier, with 97% of AUC, has been built considering 5% of outliers in the training set.

On the other hand, Table 5 demonstrates the effectiveness of PCA for high-dimensional data sets (data set 3). The best outcomes are obtained by normalizing the data by the 0/1 criterion, using an outlier factor of 5% of the training set, and selecting only 1 or 2 components. Following this approach, classifiers with AUC values above 90% can be achieved. However, using many components does not yield favorable results with this technique.

In general, it is worth noting that the computational cost of PCA is also quite low. None of the tested configurations exceeded training times of 0.6 s for the two-variable datasets, while for the 25-variable dataset, the average training times were approximately 3 s.

Finally, Table 6 and Table 7 present the solutions found with the autoencoder neural network. As with the PCA technique, the results are shown in two different tables. In addition, due to the large number of features of the third dataset, it is important to note that Table 7 only shows the best configurations for the different numbers of neurons tested.

For this method, the finest outcomes also depend on different internal computations. In the case of the first dataset (Table 6), using 1 neuron in the network hidden layer, an outlier factor of 15%, and the 0/1 normalization method; it ends with an 87.6% AUC. Nevertheless, with the second dataset (Table 6), the most promising values are attained considering 5% of the training data as outliers and also with 0/1 data normalization. Indeed, reaching up to more than 97% AUC.

On the other hand, by analyzing Table 7, it can be appreciated how the autoencoder neural networks are also capable of achieving good results in warship components with a large number of associated variables. In this case, with the third dataset, this technique reached more than 90% AUC using only 2 neurons in the hidden layer and considering a 5% outlier factor and a normalization of the data following the 0/1 criterion. However, for this dataset, it can be observed that with the increase in the number of neurons in the hidden layer, the performance is not good, and the tested classifiers do not achieve values higher than 70% AUC.

Although the results obtained in terms of AUC have been quite good with the three datasets, the computational cost of this technique is considerable, taking its toll with training times exceeding 4000 s.

Figure 5 provides a bar chart comparison of the performance of each implemented one-class technique with its respective dataset. The results are highly satisfactory, with AUC values exceeding 90%. Based on the results obtained for each technique and dataset, along with their computational costs, the k-means approach emerges as the most suitable method as it yields the highest average AUC value across all datasets analyzed.

## 6. Conclusions and Future Works

The present study has focused on the use of one-class techniques for anomaly detection in different components of a warship diesel generator. For this purpose, the performance of four one-class methods has been compared, using three different datasets corresponding to the data collected on the Spanish Navy F-100 frigates between May 2011 and May 2022.

Analyzing the achievements, it has been possible to prove that satisfactory results have been obtained in the different datasets with the four compared techniques. In the first experiment performed, with dataset 1, more than 95% AUC was recorded with the Gaussian model and k-means, while the autoencoder neural network and the PCA technique had worse performance. In the second case, dataset 2, all the implemented techniques give more than 97% AUC. Finally, with the third dataset, classifiers with an AUC higher than 90% have been achieved for all techniques except the Gaussian model.

Considering the computational cost, it has been shown that the Gaussian model, k-means, and PCA have very low rates. Moreover, the training stage of the autoencoder classifiers involves a much higher computational cost than the aforementioned techniques. In this case, the training time depends directly on the number of features of the dataset and the data normalization, given that the largest values are registered without this feature.

In general terms, this research has shown that one-class techniques are capable of detecting anomalies in the vessel component under analysis. This is a great improvement compared to the anomaly detection systems currently in operation on the ship, since their management requires very detailed knowledge of their individual functionalities of each component. Therefore, designing and tuning the current anomaly detection systems requires a large amount of expert man-hours to ensure a good performance in detecting possible failures. In addition, due to the different dynamics of the various components of a warship, a different design is required for each of these components which is an arduous task of design and adjustment.

However, with the proposal presented in this paper, the use of one-class techniques can simplify the design of these failure detection systems in diesel generator components of a warship, since no expert knowledge of the system is required. In addition, design and adjustment times could be drastically reduced, saving many thousands of Euros in commissioning these services.

On the other hand, this paper has demonstrated that one-class techniques can be used with high performance in different ship components, so the proposal’s flexibility, integration, and scalability are assured, being able to use these algorithms in other subsystems of the vessel.

In future work, the performance of these techniques on other ship components may be analyzed with the aim of obtaining a standardized approach that can detect anomalies in any component of the diesel generator. Accordingly, due to the different components within the vessel, the possibility of using hybrid classifiers based on combining clustering techniques with one-class algorithms may also be studied. In this way, more robust classifiers that could detect anomalies in various ship components would be developed. On the other hand, another line of future research will be the analysis of the anomalies detected by the proposed system in order to facilitate the management of the necessary shippable spare parts.

## Figures and Tables

**Figure 1 sensors-23-03389-f001:**
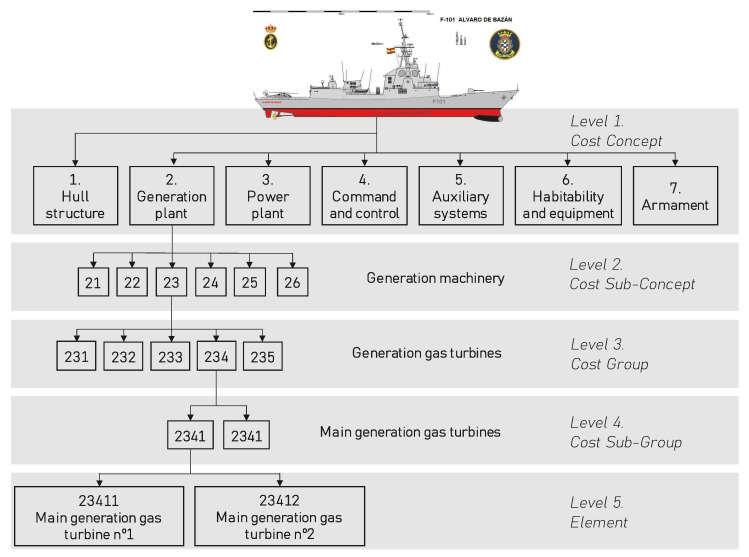
Example of the development of the functional structure [55].

**Figure 2 sensors-23-03389-f002:**
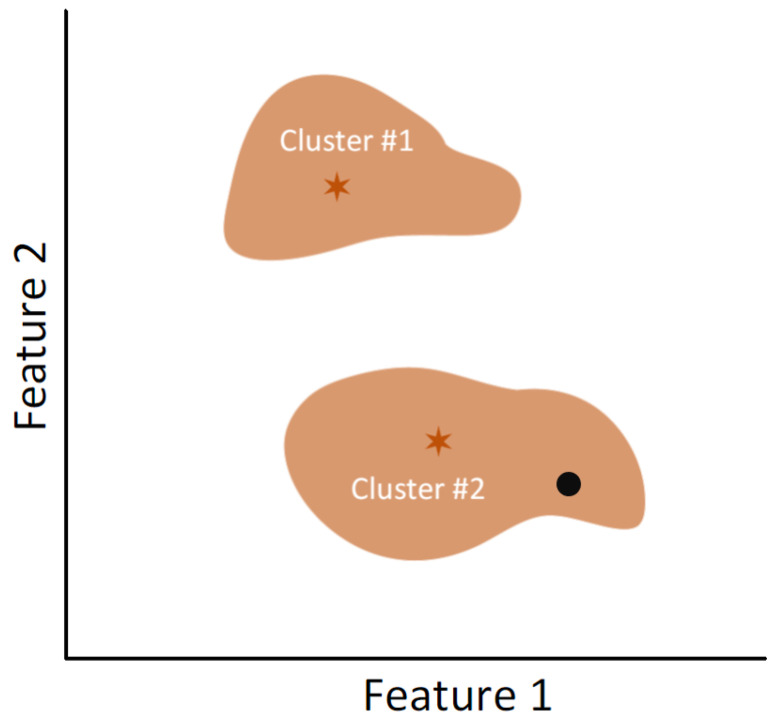
Example of k-means technique performance for two clusters.

**Figure 3 sensors-23-03389-f003:**
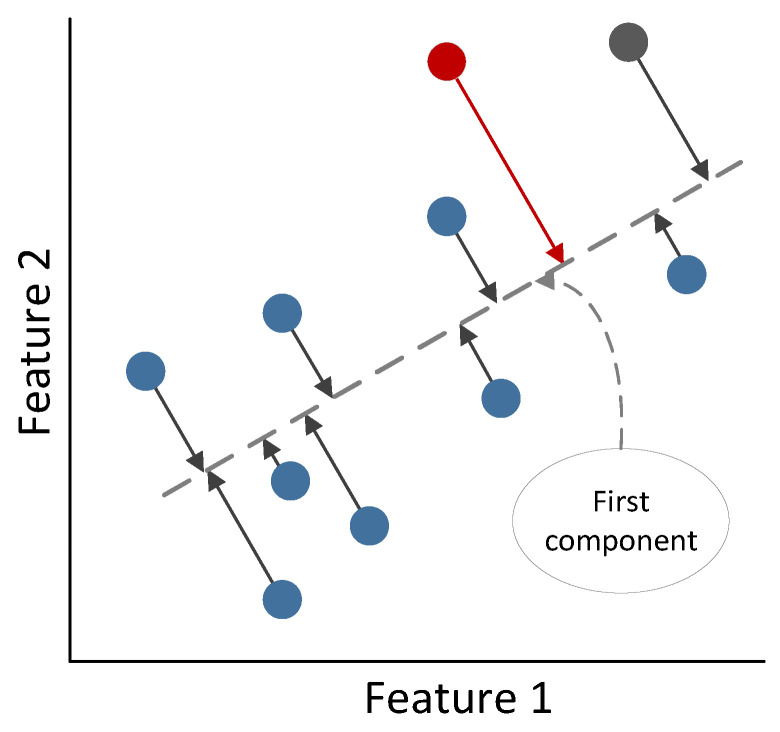
Representation of PCA for one class.

**Figure 4 sensors-23-03389-f004:**
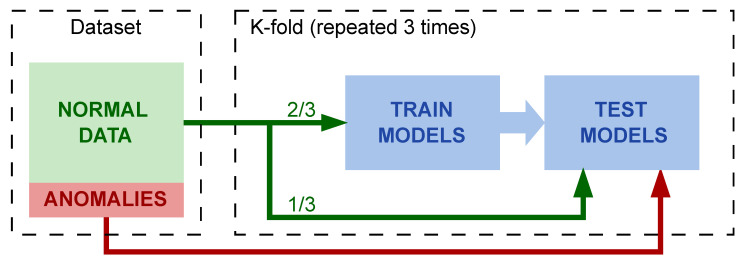
Experimental setup diagram.

**Figure 5 sensors-23-03389-f005:**
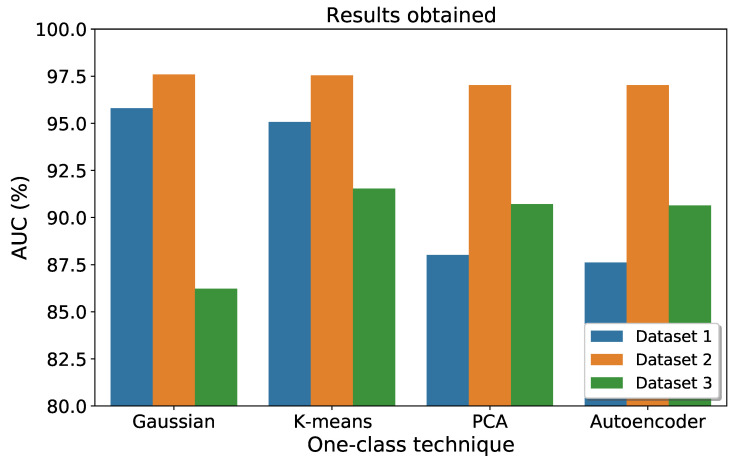
Comparison of the performance of each technique in each dataset.

**Table 1 sensors-23-03389-t001:** Configurations tested.

Evaluated Technique	Evaluated Configuration	Tested Values
Gaussian Model	Data normalization Data regularization Outlier factor (\%)	NoNorm, Norm, Zscore 0:0.003:0.009 0:5:15
K-means	Data nomalization Number of clusters Outlier factor (\%)	NoNorm, Norm, Zscore 2:2:6 0:5:15
Autoencoder	Data nomalization Neurons in the hidden layer Outlier factor (\%)	NoNorm, Norm, Zscore 1:1: $n_{var}-1$ 0:5:15
PCA	Data nomalization Number of components Outlier factor (\%)	NoNorm, Norm, Zscore 1:1: $n_{var}-1$ 0:5:15

**Table 2 sensors-23-03389-t002:** Gaussian model results.

Norm.	Regul.	Out. Factor (%)	Dataset 1		Dataset 2		Dataset 3	
			AUC (%)	T. Time (s)	AUC (%)	T. Time (s)	AUC (%)	T. Time (s)
NoNorm	0	0	50.000	0.094	50.000	0.112	50.000	0.863
NoNorm	0	5	81.317	0.082	97.501	0.089	48.365	0.937
NoNorm	0	10	81.260	0.082	95.136	0.090	56.761	0.784
NoNorm	0	15	82.460	0.072	92.507	0.086	64.647	0.921
NoNorm	0.003	0	50.000	0.077	50.000	0.087	50.000	0.874
NoNorm	0.003	5	89.071	0.077	97.537	0.091	55.218	0.936
NoNorm	0.003	10	93.209	0.079	95.019	0.090	72.345	1.056
NoNorm	0.003	15	92.529	0.076	92.565	0.092	85.718	0.773
NoNorm	0.006	0	50.000	0.079	50.000	0.082	50.000	0.965
NoNorm	0.006	5	93.676	0.073	97.584	0.079	55.942	0.748
NoNorm	0.006	10	95.048	0.074	95.065	0.082	74.366	0.967
NoNorm	0.006	15	92.528	0.076	92.562	0.082	86.222	0.946
NoNorm	0.009	0	50.000	0.077	50.000	0.075	50.000	0.796
NoNorm	0.009	5	** 95.797 **	0.076	**97.592**	0.074	56.158	1.059
NoNorm	0.009	10	95.045	0.075	95.021	0.086	76.098	0.814
NoNorm	0.009	15	92.516	0.075	92.583	0.095	**86.222**	0.739
Norm	0	0	50.000	0.074	50.000	0.083	50.000	1.085
Norm	0	5	81.384	0.078	97.501	0.093	48.366	0.920
Norm	0	10	81.254	0.076	95.136	0.101	56.688	0.928
Norm	0	15	82.453	0.078	92.507	0.095	64.648	0.770
Norm	0.003	0	50.076	0.078	50.000	0.076	50.000	0.925
Norm	0.003	5	81.313	0.074	97.501	0.089	48.150	0.819
Norm	0.003	10	81.243	0.072	95.088	0.084	53.658	0.741
Norm	0.003	15	82.469	0.085	92.511	0.100	62.551	0.839
Norm	0.006	0	50.000	0.073	50.000	0.077	50.000	0.852
Norm	0.006	5	81.376	0.074	97.504	0.077	48.150	1.051
Norm	0.006	10	81.183	0.074	95.046	0.082	54.234	0.928
Norm	0.006	15	82.473	0.072	92.504	0.087	62.771	0.839
Norm	0.009	0	50.000	0.078	50.000	0.089	50.000	0.883
Norm	0.009	5	81.307	0.078	97.515	0.086	48.149	0.883
Norm	0.009	10	81.112	0.079	95.074	0.089	54.667	0.950
Norm	0.009	15	82.488	0.078	92.501	0.094	64.002	0.822
Zscore	0	0	50.000	0.077	50.000	0.084	50.000	0.862
Zscore	0	5	81.388	0.079	97.501	0.089	48.365	0.852
Zscore	0	10	81.193	0.075	95.137	0.091	56.689	0.841
Zscore	0	15	82.467	0.074	92.506	0.091	64.649	0.951
Zscore	0.003	0	50.000	0.075	50.000	0.090	50.000	0.925
Zscore	0.003	5	81.386	0.077	97.501	0.083	48.149	0.753
Zscore	0.003	10	81.261	0.076	95.137	0.090	53.585	0.993
Zscore	0.003	15	82.477	0.080	92.531	0.090	62.556	0.946
Zscore	0.006	0	50.000	0.078	50.000	0.088	50.000	0.900
Zscore	0.006	5	81.534	0.074	97.504	0.088	48.148	0.820
Zscore	0.006	10	81.155	0.076	95.145	0.090	53.730	0.809
Zscore	0.006	15	82.478	0.077	92.514	0.094	62.340	0.958
Zscore	0.009	0	50.000	0.075	50.000	0.085	50.000	0.874
Zscore	0.009	5	81.532	0.076	97.503	0.099	48.148	0.872
Zscore	0.009	10	81.194	0.074	95.118	0.093	54.523	0.878
Zscore	0.009	15	82.470	0.078	92.517	0.082	62.267	0.865

**Table 3 sensors-23-03389-t003:** K-means results.

Norm.	N° of Clusters	Out. Factor (%)	Dataset 1		Dataset 2		Dataset 3	
			AUC (%)	T. Time (s)	AUC (%)	T. Time (s)	AUC (%)	T. Time (s)
NoNorm	2	0	50.228	0.263	50.000	0.271	50.000	4.756
NoNorm	2	5	88.435	0.229	97.509	0.273	47.716	4.473
NoNorm	2	10	91.251	0.280	95.045	0.241	61.300	4.746
NoNorm	2	15	88.648	0.304	92.559	0.278	86.572	4.692
NoNorm	4	0	50.000	0.642	50.000	0.413	50.000	6.553
NoNorm	4	5	91.340	0.529	97.520	0.398	48.004	6.977
NoNorm	4	10	89.028	0.663	95.052	0.361	73.487	5.410
NoNorm	4	15	86.670	0.593	92.540	0.514	56.844	7.174
NoNorm	6	0	50.076	0.625	50.000	0.492	50.000	7.230
NoNorm	6	5	90.155	0.650	94.489	0.670	58.692	7.526
NoNorm	6	10	84.190	0.473	95.015	0.586	45.210	7.120
NoNorm	6	15	88.033	0.910	92.515	0.630	42.706	8.634
Norm	2	0	50.000	0.271	50.000	0.300	50.000	7.109
Norm	2	5	84.373	0.353	97.524	0.314	62.796	9.290
Norm	2	10	85.206	0.326	95.040	0.340	89.068	8.206
Norm	2	15	86.559	0.334	92.559	0.328	89.902	6.341
Norm	4	0	50.000	0.569	50.000	1.018	50.000	12.225
Norm	4	5	89.866	0.540	89.801	0.937	78.969	10.016
Norm	4	10	90.562	0.843	90.328	0.600	76.912	9.586
Norm	4	15	89.376	0.599	92.522	0.637	86.655	14.039
Norm	6	0	50.000	0.802	50.000	1.123	50.000	11.278
Norm	6	5	93.246	1.094	** 97.544 **	1.223	53.423	18.574
Norm	6	10	91.043	1.130	94.989	1.107	66.796	9.461
Norm	6	15	88.726	1.232	92.517	0.770	62.852	15.000
Zscore	2	0	50.000	0.344	50.000	0.373	50.000	8.096
Zscore	2	5	85.232	0.269	97.503	0.358	78.669	14.072
Zscore	2	10	** 95.075 **	0.274	95.076	0.400	90.743	13.373
Zscore	2	15	92.559	0.281	92.553	0.330	89.685	12.033
Zscore	4	0	50.076	0.789	50.000	1.016	50.000	16.391
Zscore	4	5	95.003	0.641	97.527	0.634	90.428	17.606
Zscore	4	10	89.446	0.713	95.020	0.645	**91.539**	18.379
Zscore	4	15	90.092	1.038	92.509	1.104	89.689	15.503
Zscore	6	0	50.000	1.298	50.000	1.121	50.000	17.635
Zscore	6	5	94.997	1.076	97.530	1.268	62.290	18.634
Zscore	6	10	91.536	1.368	95.009	1.400	70.902	23.149
Zscore	6	15	90.209	1.306	92.529	1.493	74.332	20.021

**Table 4 sensors-23-03389-t004:** PCA results for dataset 1 and 2.

Norm.	Comp.	Out. Factor (%)	Dataset 1		Dataset 2	
			AUC (%)	T. Time (s)	AUC (%)	T. Time (s)
NoNorm	1	0	50.000	0.484	50.000	0.488
NoNorm	1	5	59.834	0.518	48.026	0.459
NoNorm	1	10	58.336	0.486	46.032	0.439
NoNorm	1	15	55.814	0.470	44.536	0.441
Norm	1	0	50.000	0.457	50.000	0.461
Norm	1	5	85.668	0.523	** 97.027 **	0.438
Norm	1	10	86.524	0.565	95.021	0.451
Norm	1	15	** 88.009 **	0.484	92.552	0.478
Zscore	1	0	50.000	0.462	50.000	0.441
Zscore	1	5	70.969	0.458	69.237	0.440
Zscore	1	10	69.865	0.507	76.894	0.441
Zscore	1	15	68.618	0.484	80.425	0.460

**Table 5 sensors-23-03389-t005:** PCA results for dataset 3.

Norm.	Comp.	Out. Factor (%)	Dataset 3	
			AUC (%)	T. Time (s)
Norm	1	5	90.357	2.603
Norm	2	5	**90.717**	3.353
NoNorm	3	15	72.586	3.045
Norm	4	15	53.828	3.138
NoNorm	5	15	70.494	3.497
NoNorm	6	15	75.689	3.229
NoNorm	7	15	59.308	2.679
NoNorm	8	0	50.000	3.126
NoNorm	9	15	50.148	2.608
NoNorm	10	0	50.000	3.182
NoNorm	11	15	51.809	3.233
NoNorm	12	15	57.290	2.819
Norm	13	15	52.603	3.083
Norm	14	15	58.655	3.468
Zscore	15	15	59.890	3.121
Zscore	16	15	63.352	2.979
Zscore	17	15	63.352	3.097
NoNorm	18	5	53.130	3.132
Norm	19	15	54.117	3.030
Norm	20	15	61.113	3.461
Zscore	21	15	63.493	2.766
Norm	22	15	66.233	3.058
NoNorm	23	15	54.984	3.177
Zscore	24	15	62.700	3.105

**Table 6 sensors-23-03389-t006:** Autoencoder results for dataset 1 and 2.

Norm.	N° Neurons	Out. Factor (%)	Dataset 1		Dataset 2	
			AUC (%)	T. Time (s)	AUC (%)	T. Time (s)
NoNorm	1	0	50.000	357.132	50.000	147.926
NoNorm	1	5	84.857	352.384	48.031	157.925
NoNorm	1	10	71.600	189.100	46.035	197.221
NoNorm	1	15	79.714	285.890	44.568	128.557
Norm	1	0	50.000	30.100	50.000	31.605
Norm	1	5	85.826	32.068	**97.026**	45.150
Norm	1	10	86.513	39.528	95.020	55.993
Norm	1	15	**87.617**	41.665	92.550	41.893
Zscore	1	0	50.000	34.112	50.000	21.486
Zscore	1	5	70.883	21.553	69.231	32.899
Zscore	1	10	70.133	13.587	76.874	50.331
Zscore	1	15	68.329	14.634	81.100	27.728

**Table 7 sensors-23-03389-t007:** Autoencoder results for dataset 3.

Norm.	N° Neurons	Out. Factor (%)	Dataset 3	
			AUC (%)	T. Time (s)
Norm	1	5	90.358	578.872
Norm	2	5	**90.646**	746.915
Zscore	3	15	71.300	699.515
Zscore	4	5	54.426	957.287
Zscore	5	15	52.447	1787.564
Zscore	6	0	50.000	1621.264
Norm	7	15	75.036	2483.040
Zscore	8	15	64.628	2248.230
Zscore	9	0	50.000	2473.127
Zscore	10	0	50.000	2539.572
Zscore	11	0	50.000	2636.626
Zscore	12	0	50.000	2682.045
Norm	13	15	57.131	3221.705
Zscore	14	15	57.675	2962.068
Zscore	15	15	56.630	3085.607
Norm	16	15	54.603	3490.254
Norm	17	15	56.859	3835.607
Norm	18	15	59.668	3910.105
Zscore	19	15	61.986	3670.120
Zscore	20	15	62.389	3764.557
Zscore	21	15	61.268	3992.231
Norm	22	15	64.799	3722.349
Norm	23	15	62.389	4018.956
Norm	24	15	60.247	4125.477

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
