# Peer review of "A Fault-Detection System Approach for the Optimization of Warship Equipment Replacement Parts Based on Operation Parameters"

_sensors, 2023, doi:10.3390/s23073389_

Round 1
Reviewer 1 Report
In this manuscript, the authors suggested a comparison between different one-class techniques, such as statistical models, geometric boundaries, or dimensional reduction, to face anomaly detection in specific subsystems of a warship, with the prospect of applying it to the whole ship. Generally, the submitted manuscript has some merits and is of practical significance for the operation and maintenance of the warships. However, the manuscript is not well-written in the current version. The following comments can be considered to improve the quality.
1. Abstract
The abstract is a little logic mess. Please rewrite the abstract and further clarify the concerned research gaps in the existing systems, the proposed approach, and the novelties in a logical manner.
2. Introduction
The irrelevant research backgrounds with the concerned topic can be eliminated, such as the first six paragraphs, please refine this part. The existing writing style is somewhat like a general scientifical article, not a research manuscript. In addition, please summarize the research gaps in the existing studies and further clarify the research contributions in a item-by-item manner. In the current version, it is hard to quickly grasp the real contributions of this manuscript.
3. Introduction
The literature review is a little limited and not profound. The existing latest research achievements of the predictive maintenance system implementation issues under limited labeled samples and intelligent fault prognostic issues (10.1016/j.ymssp.2022.109628) should also be reviewed.
4. Materials and Methods
Please present some fundamental basics of the employed methods, such as the concerned equations and demonstrative diagrams. Only brief introductions of some methods are included in the existing manuscript.
5. Results
It is also suggested to add some explainable figures to further illustrate the experimental results, only some tables are included. In addition, the results from Gaussian model, K-means, and PCA can be compared in a same baseline ?
6. About the writing of this paper, grammar errors and typos need to be carefully checked and amended.
Reviewer 2 Report
Conclusion: In view of the need for fault detection system in the whole life cycle management of complex and expensive systems such as ships, this paper studies the fault diagnosis method of ship systems based on the ship operation parameters. The accuracy of the method is verified by using the actual operation data of the Spanish Navy F-100 frigate and taking the ship diesel generator as an example, which can lay the foundation for the subsequent fault diagnosis research of the whole ship and has certain engineering significance. However, targeted modifications are needed, and the specific comments are as follows:
1. It is suggested to modify the Abstract, focusing on the content, function and application effect of the fault detection system in this paper.
2. The Introduction part of the article contains a lot of content, it is suggested to reduce it appropriately,and highlight the research background of this article.
3. In the section of related research, the author lists a large number of fault diagnosis methods, involving different fields and objects. It is suggested to take fault diagnosis methods based on operating parameters and ship systems as objects, discuss the relevant research progress, and highlight the innovation of this paper.
4. The conclusion part is suggested to further discuss the research tasks and achievements of this paper, highlight the advantages of the research work and the application prospects.
5. The article needs to further strengthen the experimental verification. It is suggested to carry out experimental verification for a subsystem or add cases of other systems to verify the correctness and advantages of the method.
